# Assessment of Photoactivated Chlorophyllin Production of Singlet Oxygen and Inactivation of Foodborne Pathogens

Cristina Pablos [1], Javier Marugán [1,2,*], Rafael van Grieken [1], Jeremy W. J. Hamilton [3], Nigel G. Ternan [4] and Patrick S. M. Dunlop [3,*]

[1] Department of Chemical and Environmental Technology, ESCET, Universidad Rey Juan Carlos, c/Tulipán s/n, 28933 Móstoles, Madrid, Spain; cristina.pablos@urjc.es (C.P.); rafael.vangrieken@urjc.es (R.v.G.)

[2] Instituto de Tecnologías para la Sostenibilidad, Universidad Rey Juan Carlos, C/Tulipán s/n, 28933 Móstoles, Madrid, Spain

[3] Nanotechnology and Integrated BioEngineering Centre (NIBEC), Ulster University, Belfast BT15 1AP, UK; jwj.hamilton@ulster.ac.uk

[4] School of Biomedical Sciences, Ulster University, Coleraine, Londonderry BT52 1SA, UK; ng.ternan@ulster.ac.uk

* Correspondence: javier.marugan@urjc.es (J.M.); psm.dunlop@ulster.ac.uk (P.S.M.D.)

**Abstract:** Singlet oxygen ($^1O_2$) is known to have antibacterial activity; however, production can involve complex processes with expensive chemical precursors and/or significant energy input. Recent studies have confirmed the generation of $^1O_2$ through the activation of photosensitizer molecules (PSs) with visible light in the presence of oxygen. Given the increase in the incidence of foodborne diseases associated with cross-contamination in food-processing industries, which is becoming a major concern, food-safe additives, such as chlorophyllins, have been studied for their ability to act as PSs. The fluorescent probe Singlet Oxygen Sensor Green (SOSG®) was used to estimate $^1O_2$ formation upon the irradiation of traditional PSs (rose bengal (RB), chlorin 6 (ce6)) and novel chlorophyllins, sodium magnesium (NaChl) and sodium copper (NaCuChl), with both simulated-solar and visible light. NaChl gave rise to a similar $^1O_2$ production rate when compared to RB and ce6. Basic mixing was shown to introduce sufficient oxygen to the PS solutions, preventing the limitation of the $^1O_2$ production rate. The NaChl-based inactivation of Gram-positive *S. aureus* and Gram-negative *E. coli* was demonstrated with a 5-log reduction with UV–Vis light. The NaChl-based inactivation of Gram-positive *S. aureus* was accomplished with a 2-log reduction after 105 min of visible-light irradiation and a 3-log reduction following 150 min of exposure from an initial viable bacterial concentration of $10^6$ CFU mL$^{-1}$. CHS-NaChl-based photosensitization under visible light enhanced Gram-negative *E. coli* inactivation and provided a strong bacteriostatic effect preventing *E. coli* proliferation. The difference in the ability of NaChl and CHS-NaChl complexes to inactivate Gram-positive and Gram-negative bacteria was confirmed to result from the cell wall structure, which impacted PS–bacteria attachment and therefore the production of localized singlet oxygen.

**Keywords:** chlorophyllin; singlet oxygen; Gram-positive; Gram-negative; food safety; photosensitization

## 1. Introduction

According to recent reports from the Word Health Organization (WHO), the incidence of foodborne disease associated with microbial pathogens is widespread and of significant public health concern in both developed and developing countries [1]. In addition to viral pathogens, *Salmonella* sp., *Campylobacter* sp., and *Escherichia coli*, Gram-negative bacteria, together with *Listeria* sp. and *Staphylococcus* sp., Gram-positive bacteria, are reported to be the most prevalent foodborne pathogens in many countries [2]. The contamination of food can arise through contact with operatives, surfaces, and equipment during processing—usually termed cross-contamination—which results in microbial spoilage, a reduction in product shelf-life, and post-consumption foodborne illness. Therefore, in addition to the public

health impact, the microbial contamination of food has significant economic and environmental impacts and has become a key challenge for the food-processing industry. Significant concerns have also been raised in relation to antimicrobial-resistant (AMR) pathogens in the food chain, with ongoing research attempting to elucidate human and animal health risks [3].

In many sectors, microbial contamination is commonly addressed via post-processing thermal sterilization techniques, such as pasteurization, canning, roasting, or frying (often to produce "ready to eat" products), but for some products thermal treatment may affect the nutritional properties in addition to the texture, taste, and smell [4]. With respect to cross-contamination via food-processing surfaces and equipment, thermal processes can be used in combination with detergent washing to render surfaces pathogen free, and whilst the addition of disinfectants such as NaOCl, ozone, electrolyzed water, etc., may enhance efficiency, their use can lead to the formation of potentially harmful chloro-organic disinfection by-products (DBPs), with carcinogenic and mutagenic effects on mammals [5,6]. Microbial resistance to common disinfectants has also been reported, with evidence of the potential to exacerbate AMR [7]. Non-thermal methods with which to inactivate microbial pathogens and biofilms on surfaces include atmospheric pressure plasma and ultrasound, along with the implementation of antimicrobial surfaces and pulsed as well as laser light irradiation. Although proven at the laboratory scale, these alternative approaches often involve specialized equipment, highly trained personnel, and high capital, operational, and maintenance costs, as well as requiring longer processing times than conventional approaches [8]; therefore, considerable efforts are underway to develop effective technologies to prevent cross-contamination within food-processing environments using more sustainable and environmentally friendly approaches with minimal energy and chemical input.

Antimicrobial photodynamic therapy (APDT) is a promising disinfection technique with low energy demand, which is primarily used as a treatment for localized cancers [9]. The process is based on the use of visible light of an appropriate wavelength to excite dyes, known as photosensitizer molecules (PSs), from a low-energy ground state to a higher-energy state, often in the presence of oxygen. Photon absorption results in energy transfer via two possible pathways (so-called Type I and Type II pathways), leading to the formation of reactive oxygen species (ROSs) such as hydroxyl ($OH^{\bullet}$), superoxide anion radicals ($O_2^{-\bullet}$), and singlet oxygen ($^1O_2$) [10–13]. ROSs can oxidize many biological molecules and render pathogens inactive via several mechanisms. The PSs chosen to dictate the photochemistry mechanism and as such the ROSs generated; the excitation of the most common PSs results in Type II pathway chemistry and the production of singlet oxygen [14]. In a Type II pathway, the photosensitizer directly transfers energy to oxygen, forming reactive singlet oxygen. Subsequently, singlet oxygen interacts with biomolecules in the surrounding environment. High-oxygen-concentration systems are more likely to use a Type II mechanism, while a Type I mechanism is predominant in oxygen-depleted settings [15–17]. Singlet oxygen possesses significant antimicrobial activity; thus, PS compounds such as rose bengal (RB), methylene blue (MB), and chlorin e6 (ce6) are commonly used as photosensitizers in APDT due to their high singlet oxygen quantum yields (0.76, 0.5, and 0.7, respectively) [11,13,18–20].

APDT has been shown to effectively inactivate a wide range of microorganisms, including viruses, yeasts, spores, and both Gram-negative as well as Gram-positive bacteria, despite the former being much more challenging to inactivate [19,21–25]. It has been reported that the use of cationic PSs or positively charged binding molecules can permit close contact between PSs and bacterial outer structures, and therefore result in the production of singlet oxygen at the outer surface of the bacteria, resulting in enhanced rates of inactivation [19,23–27]. To further enhance the performance of PSs, the use of chitosan (CHS)—a derivative of chitin sourced from the shell waste of crustacean-processing industries and accepted as a food additive [28]—is gaining attention as a cationic binding molecule in APDT. CHS not only improves bacterial–PS interaction [13,25] but also possesses antimicrobial properties, with efficacy reported against both Gram-positive and Gram-negative

bacteria [29]. The primary attraction mechanism for CHS relates to the presence of $NH_3^+$ group interactions with negatively charged bacterial cell surfaces, whereby subsequent membrane rupture leads to the leakage of intracellular constituents from the bacterium and inactivation [29].

Given the potential for the excitation of PSs by visible light, there is significant interest in APDT as a non-thermal, low-cost, and environmentally friendly alternative process to reduce bacterial contamination on food production surfaces. From a practical point of view, the application of PSs via pre-coating onto surfaces, or by spraying PSs onto equipment such as conveyor belts, etc., offers labor-free delivery, reducing capital and operating costs [5,24,30,31]. Porphyrins such as chlorophyll and its derivatives have extensively been reported as PSs in APDT [21]; however, only few publications examine the applications for food products, food-related surfaces, and food-packaging materials [6,21,32–35]. In previous work using chlorophyllin, photosensitization systems usually required dark pre-incubation with target bacteria of between 2–120 min, prior to a second period for light-driven inactivation [36,37]. This dual-exposure protocol unfortunately limits the practical application of this promising disinfection approach.

Photosensitizers used for food applications must not show detrimental effects on either the appearance or organoleptic properties of food products. Chlorophyllins such as magnesium chlorophyllin (NaChl) and copper chlorophyllin (NaChlCu) are generally regarded as safe food additives (E-140 and E-141, respectively), and are permitted as food colorants according to EC regulations [38]—as such, they form ideal candidates for APDT in food-processing environments [5,6,20,21,30].

The aim of this work was to evaluate the biocidal and biostatic effects of food-safe chlorophyllin compounds for application in food-processing environments using low-energy photon sources. To ensure ideal conditions for high levels of pathogen inactivation, the project aimed to gain an understanding of the environment required to generate a high concentration of $^1O_2$—using a Singlet Oxygen Sensor Green (SOSG$^®$) fluorescence probe to detect $^1O_2$—with minimal requirements for preparation, the use of expensive reagents, or complex approaches.

## 2. Results and Discussion

### 2.1. Singlet Oxygen Production by Photosensitization

#### 2.1.1. Influence of Photosensitizer Concentration on $^1O_2$ Production

The absorption spectra for each photosensitizer were confirmed by spectroscopy, with peak absorption noted at 535, 403, 405, and 401 nm for RB, ce6, NaChl, and NaCuChl, respectively (Figure S1), in agreement with the works of Luksiene and Brovko [21] for RB and ce6, and that of Phasupan et al. [39] for NaChl and NaCuChl. Figure S1 also shows the output spectra of the photon sources (UV–Vis component for both simulated-solar and CFL lamps), demonstrating the matching of source to PS excitation across a few key regions in the visible region.

Figure 1 shows $^1O_2$ generation due to the irradiation of photosensitizer compounds at an initial concentration of 0.5 µM. No significant fluorescence was observed from the SOSG probe in the dark, nor when either SOSG or the photosensitizer were irradiated alone. Therefore, it can be concluded that the fluorescence enhancement results from singlet oxygen generation due to the photoexcitation of each PS.

In the presence of $^1O_2$, SOSG can react with $^1O_2$ to produce SOSG endoperoxides (SOSG-EP) with strong fluorescence [40]. A linear increase in green fluorescence enhancement (F), due to the formation of SOSG-EP, was observed as a function of time for up to 16 min of irradiation. Afterwards, this increase is not linear with time. This saturation effect at higher concentrations has been previously reported [40,41]. The sustained production of high concentrations of singlet oxygen from all PSs was observed for at least 60 min. Levels of $^1O_2$ produced from NaChl were comparable to those of RB and ce6—demonstrating the biocidal potential for the implementation of the food-safe compound. NaCuChl provided the lowest levels of singlet oxygen production. As the lifetime of the excited PS and gener-

ated singlet oxygen is within the microsecond range [11,21], it is important to ensure the proximity of the PS to the bacterium to ensure the potential for oxidation and inactivation.

The effect of the PS concentration in the range 0.01–1.0 µM on singlet oxygen generation was measured with a fixed concentration of SOSG in excess (2 µM). The zero-order kinetic constants were calculated from the initial reaction rate of the time-dependent fluorescence enhancement of SOSG, i.e., the concentration of SOSG-EP. As reported in the literature, $^{1}O_2$ production rates are often observed to increase linearly as a function of the initial PS concentration; however, in this case we cannot report that expected effect due to the narrow range of PS concentration. Within the range of the concentrations evaluated, we obtained maximum singlet oxygen formation at 0.5 µM, but recognize the potential for rate limitation at 1 µM of PSs due to the fixed concentration of SOSG (2 µM). As such, 0.5 µM was considered as an appropriate PS concentration for further experimentation, effectively using the incident radiation to produce consistent levels of singlet oxygen.

Both chlorophyllin-based compounds exhibited singlet oxygen production under "practical" conditions, whereas the literature typically reports the use of concentration values ranging from 10 to 150 µM and exposure to higher light intensities [26,37,42,43]. While the initial $^{1}O_2$ rate constant for NaChl ($293 \pm 22$ µM·min$^{-1}$) was similar ($p > 0.05$) to those obtained for RB and ce6, 323 and $291 \pm 21$ min$^{-1}$, respectively, NaCuChl showed a lower value of $143 \pm 23$ min$^{-1}$ ($p \leq 0.05$), demonstrating significant potential for the use of NaChl as an effective food-safe biocidal compound. We may have expected the highest singlet oxygen production for RB and ce6, with values of reported quantum yield of 0.75 and 0.70, respectively [21,39], followed by NaChl, with a reported quantum yield of 0.39. Low yields of singlet oxygen production are consistent with reported quantum yields for NaCuChl of 0.03 [39].

To ensure that singlet oxygen was the only major reactive oxygen species generated by the chlorophyllin-based compounds, the production of a superoxide anion radical was evaluated throughout the photoirradiation of NaChl (initial concentration of PSs = 0.5 µM). The use of a colorimetric XTT formazan formation-based probe confirmed no detection of $O_2^{-\bullet}$. Hence, we have confidence that $^{1}O_2$ is the only ROS generated via the irradiation of chlorophyllins by UV–Vis photons.

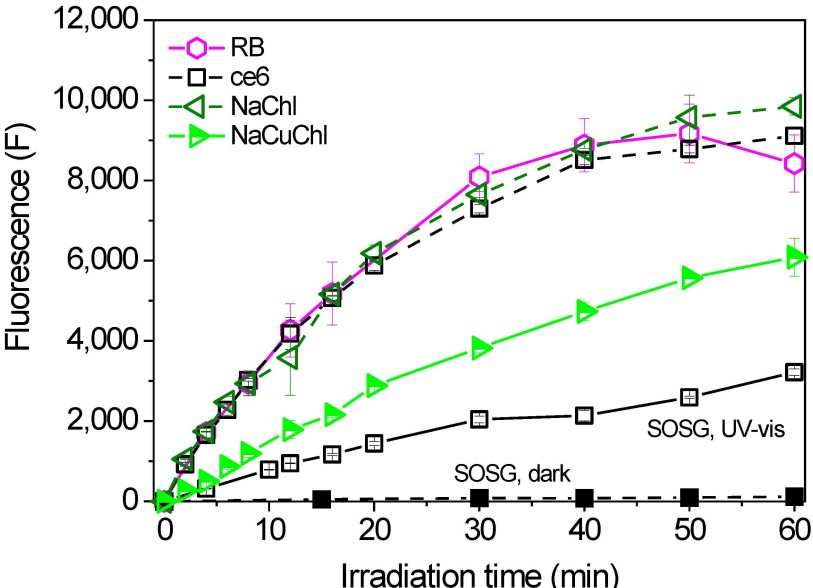

**Figure 1.** Green fluorescence enhancement due to $^{1}O_2$ generation arising from the photoirradiation of a 0.5 µM PS solution in deionized water and 2 µM SOSG under UV–Vis radiation.

### 2.1.2. Effect of Practical Considerations on $^1O_2$ Production, (a) Solution pH

Previous studies report the need to optimize the initial solution pH to ensure effective and continued singlet oxygen production; however, there is no clear consensus, with authors reporting pH values of less than 7 right up to approximately pH 10 [13,14,31,42–44]. Different pH values may lead to changes either in the chemical structure of a PS, shifting its maximum peak of absorption, or protonation, resulting in different singlet oxygen formation quantum yields.

Figure 2 shows the influence of solution pH on the generation of singlet oxygen upon exposure to UV–Vis irradiation, with maximum production observed at the unmodified solution pH of 5.

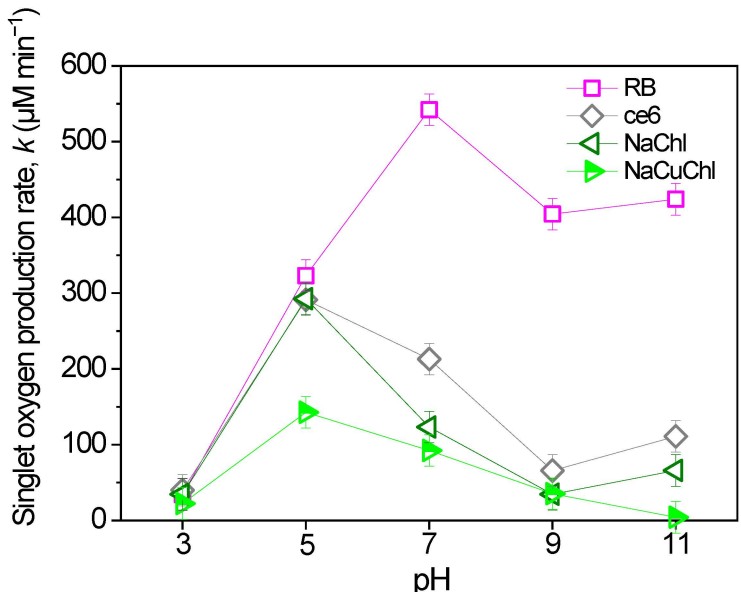

**Figure 2.** Singlet oxygen production rate as a function of initial solution pH during irradiation with UV–Vis photons. Photosensitizer initial concentration = 0.5 μM.

A decrease in the rate of singlet oxygen production was observed at neutral and basic pH levels (7, 9, and 11), and pH 3 also resulted in a significant reduction in $k$ ($p \leq 0.05$) from the unmodified solution pH. RB potential to produce singlet oxygen was only affected at low pH values, in agreement with Neckers [45], where acid conditions result in the formation of an ester group causing ring closure. Thus, at a pH range of 2.7–3.2, an important shift occurs in the absorption maximum of RB. When pH is less acidic, the open-ring RB structure dominates [46]. Moreover, Lin et al. [40] noted the reaction between SOSG and singlet oxygen to be sensitive to the solution pH, temperature, and dissolved oxygen concentration, and as such further investigation could be performed to supplement the use of this indirect $^1O_2$ probe in specific microenvironments. From a practical perspective, the data confirm that a general supply of distilled water (slightly acidic) would be sufficient for the preparation of a functional chlorophyllin solution for industrial application.

### 2.1.3. Effect of Practical Considerations on $^1O_2$ Production, (b) Oxygen Concentration

Increasing the concentration of oxygen within the system did not result in additional singlet oxygen generation for ce6, NaChl, and NaCuChl solutions (Figure 3); however, enhanced $^1O_2$ generation was observed for RB. It is generally accepted that the presence of oxygen is necessary in the system to yield singlet oxygen [44,47–50] specifically in Type II reactions [51]; however, full oxygen saturation is not required and the additional sparging could induce PS agglomeration, impacting RB to a lesser extent than the other PSs.

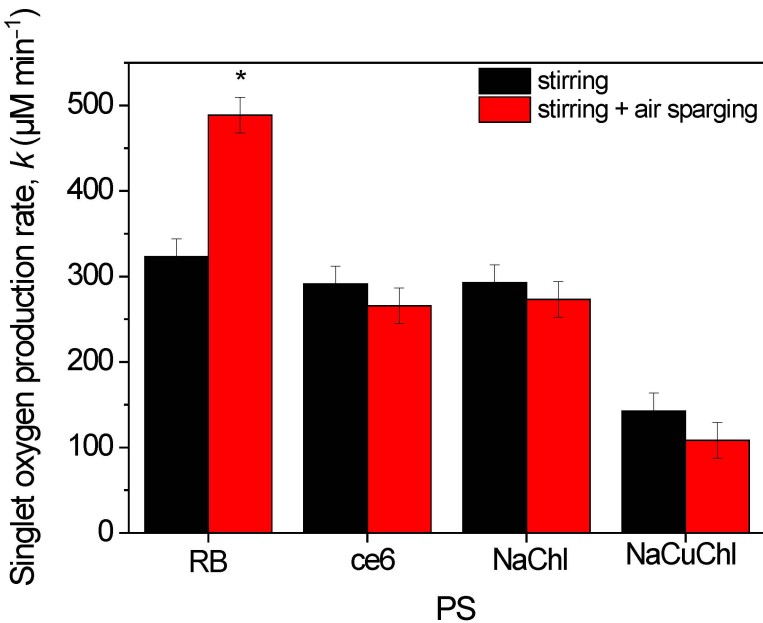

**Figure 3.** Effect of the increase in oxygen in the reaction solution on the singlet oxygen production rate in deionized water throughout the irradiation with UV–Vis light of different photosensitizers at 0.5 μM as the initial concentration. Significant differences ($p \leq 0.05$) have been indicated with an asterisk.

Whilst the data support the need for the presence of oxygen, simple agitation (via stirring) was demonstrated to be sufficient to yield good production of singlet oxygen with chlorophyllin-based systems, again demonstrating the applicability of the system for industrial applications.

### 2.2. Inactivation of Gram-Positive and Gram-Negative Bacteria Using Chlorophyllin-Based Photosensitization

#### 2.2.1. Bacterial Inactivation under UV–Vis Irradiation

High levels of *S. aureus* inactivation were observed (5-log reduction) upon UV–Vis irradiation of 0.5 μM NaChl (Figure 4). In the absence of light (dark control), bacterial inactivation was not observed. Photoinactivation was observed in the absence of NaChl, where UV components of the photon source resulted in bacterial inactivation over an extended period; the addition of NaChl was observed to halve the exposure time required to attain 5-log inactivation.

Photolysis (i.e., direct inactivation arising from exposure to UV–Vis energy) of both Gram-positive and Gram-negative organisms is widely reported, with energy in the UV-A region attributed as the primary active agent [52–54]. The photosensitivity of *E. coli* to UV was also demonstrated in Figure 4 (inset) in comparison with negligible additional inactivation shown upon the addition of NaChl (Figure 4 inset and Figure S2). Greater levels of photoinactivation were observed for Gram-negative in comparison with Gram-positive bacteria, and, despite considerable research, the mechanism to explain this effect is not still clear. Differences in cell wall structure have been described as the primary drivers for the photosensitivity of Gram-negative bacteria, with several authors describing (i) differing physiological changes induced by irradiation; (ii) different response mechanisms to UV-B and UV-A light as consequences of the upregulation of protection functions, such as the generation of extracellular polymeric substances (EPSs) as a response to sublethal stress, coupled with photo-repair mechanisms, including recA; and the (iii) modification of the membrane structure with thicker cell walls observed within Gram-positive bacteria upon exposure to simulated-solar light [52–55].

The photoactivation of 0.5 μM NaChl resulted in the increased bacterial inactivation of *S. aureus* when compared to that for *E. coli*. Gram-positive bacteria have been

reported to contain a more porous cell wall than the outer double membrane of Gram-negative species [18,19,30,34], and, as a result, singlet oxygen [30] and PSs [13,23,26,36,56] can transfer across/through the Gram-positive cell structure, increasing bacterial inactivation efficiency. Additionally, the increased production of endogenous porphyrins by Gram-positive bacteria [31] may result in higher sensitivity to photosensitization-based processes [12,18,57]. In addition, anionic PSs, such as chlorophyllins, have been reported to bind to Gram-positive bacteria, despite the negative surface charge, more favorably than to Gram-negative species. Thus, when the PSs are activated and singlet oxygen formed, ROSs are generated near Gram-positive bacteria. Several studies have confirmed that a successful alternative approach to improve Gram-negative bacteria inactivation might be by either the implementation of positively charged photosensitizers [13,27,58] or positively charged molecules mediating PS–bacteria interaction [13,23,27,34,59]. In addition, cationic molecules or conjugates have also been reported to increase the permeability of the outer membrane of Gram-negative bacteria, since they extract divalent cations from the LPS [23,27], favoring the diffusion of PSs through the outer membrane. In this work, we did not observe additional inactivation under UV–Vis irradiation with the addition of CHS to the chlorophyllin solution for either *E. coli* or *S. aureus.*, and agree with the findings of Buchovec et al., 2017 [59], and López-Carballo et al., 2018 [57], that further work is required to fully elucidate the mechanism of action.

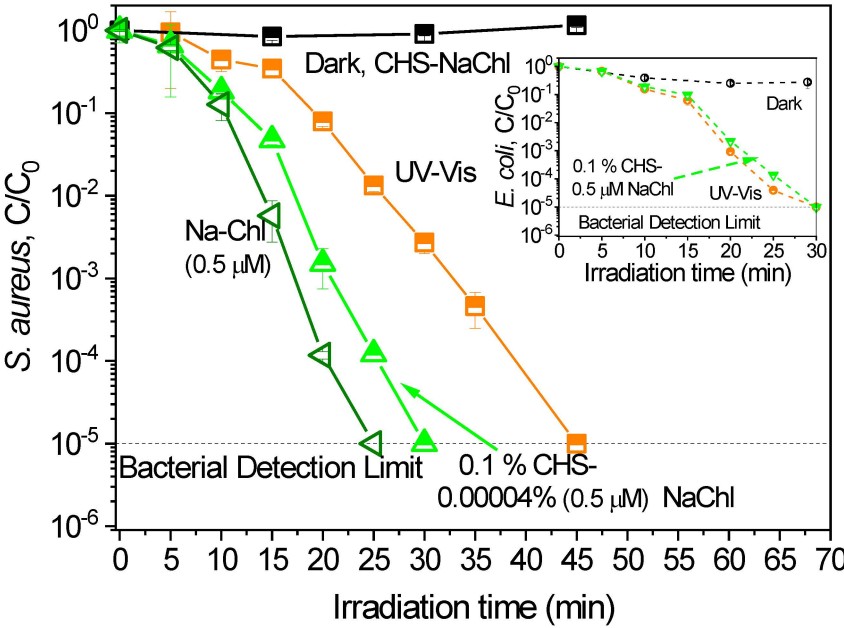

**Figure 4.** Inactivation of *S. aureus* and *E. coli* (figure inset) by chlorophyllin-based photosensitization in ¼ strength Ringer's solution under UV–Vis light. PSs: 0.5 μM NaChl, and the complex 0.1% CHS-NaChl 0.00004% (0.5 μM).

2.2.2. Bacterial Inactivation under Visible Light

With respect to industrial applications, irradiation with standard indoor light would be more practical than a solar-simulated source, and, as such, experiments were carried out under visible light using a CFL lamp. The photosensitized-based inactivation of a high concentration of *S. aureus* was confirmed in the presence of NaChl (3-log reduction), although the lower-energy source required additional exposure time (Figure 5a). No reduction in viability was observed upon the exposure of *S. aureus* to the CFL source (Figure 5a). NaCuChl compounds were not effective upon exposure to the visible-only photon source (Figure 5b). The higher efficiency of NaChl compared to that of NaCuChl was previously reported [30,57]; however, conversely, Luksiene and Paskeviciute [5] observed reasonable levels of inactivation with NaCuChl. Additionally, Josewin et al. reported the

inactivation of *Salmonella* at 4 and 20 °C using 405 nm irradiation; however, there was no significant ($p \geq 0.05$) difference in log reduction between 460 nm illumination alone and NaCuChl-mediated LED illumination for the inactivation of *Listeria monocytogenes* at temperatures of 4 and 20 °C, nor for *Salmonella* at 20 °C [60]. Low levels of bacterial inactivation with NaCuChl correspond to low singlet oxygen production, as demonstrated in Figure 1, the kinetic constant values reported previously, and the low quantum yield reported for NaCuChl, 0.03 [39].

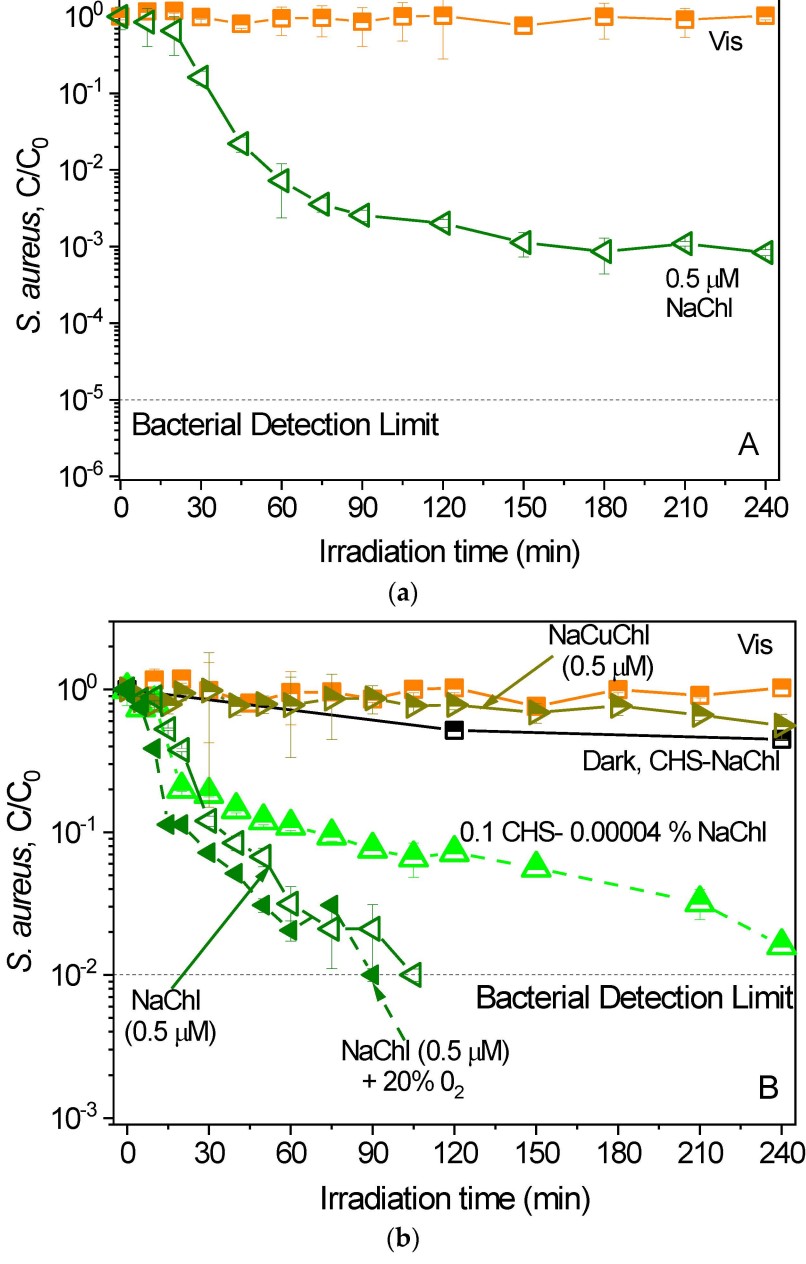

**Figure 5.** Inactivation of *S. aureus* by chlorophyllin-based photosensitization in ¼ strength Ringer's solution under visible light: (**a**) PS: 0.5 µM NaChl. Bacterial initial concentration: $10^6$ CFU·mL$^{-1}$. (**b**) PSs: 0.5 µM NaChl and 0.5 µM NaChl under a high concentration of oxygen corresponding to an increase of 20%; 0.5 µM NaCuChl; and the complex CHS (0.1%)-NaChl (0.00004%, ca. 0.5 µM). Bacterial initial concentration: $10^3$ CFU·mL$^{-1}$.

No additional effect was observed with the complex of CHS-NaChl (Figure 5b). Whilst Huang et al. (2012) [19] reported cationic CHS as binding more strongly to Gram-negative

bacteria than Gram-positive species, other groups disagree. It is accepted that CHS possesses antimicrobial effects arising from the increased permeability of the cell membrane for both Gram-positive and Gram-negative bacteria [28–30], with increased antimicrobial properties against Gram-positive bacteria arising from the alteration of the outer membrane of Gram-positive species [29]. George et al. [23] confirmed the uptake of anionic PSs coupled with divalent cations, which enhanced Gram-positive and Gram-negative inactivation rates.

In the present work, a complex, CHS-NaChl, was studied for the photosensitized inactivation of Gram-positive and Gram-negative bacteria (Figures 5b and 6, respectively). In agreement with other studies, Gram-negative bacteria (*E. coli*) inactivation was enhanced by the CHS couple. Similar effects relating to photosensitization-based inactivation with conjugates of anionic PSs and positively charged CHS have been previously observed [20,25,26,30,31]. On the one hand, the ability of CHS to bind to the outer membrane was reported to increase permeability and the uptake of anionic PSs by the cell [30], with the leakage of intracellular constituents observed by Liu et al. [29]; however, on the other hand, water-soluble CHS–anionic PS complexes present positive surface charge at pH values lower than 6.3 [29] due to $NH_3^+$ groups present in CHS [21,29], which permits binding to negative groups on the outer structures of the bacterial cell surface, neutralizing anionic charges, reducing the repulsion between anionic PSs and negative charges on the bacterial cell wall. Thus, the attachment of PSs to bacteria has been reported to be enhanced by the addition of CHS, and, as such, the generation of singlet oxygen near the cell surface [5].

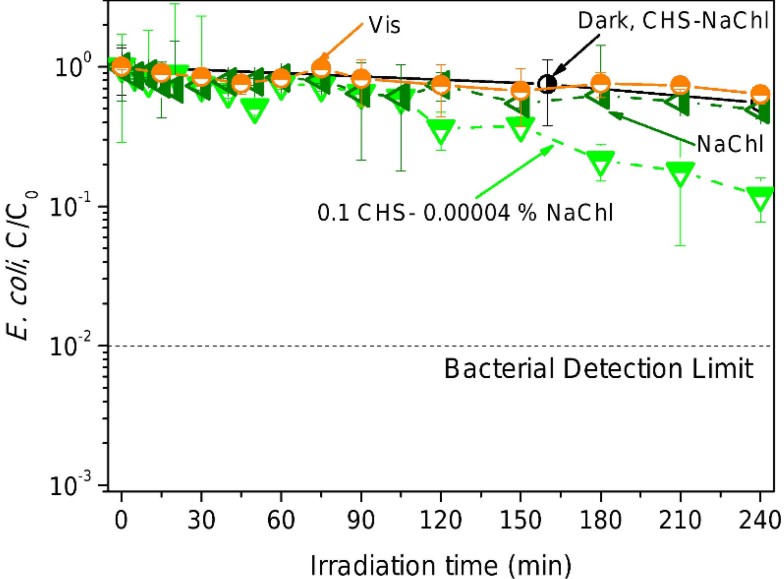

**Figure 6.** Inactivation of *E. coli* by chlorophyllin-based photosensitization in ¼ strength Ringer's solution under visible light. PSs: 0.5 µM NaChl and the complex CHS (0.1%)-NaChl (0.00004%, ca. 0.5 µM).

However, no increase in inactivation was observed for Gram-positive *S. aureus*. A potential explanation may relate to the size of chitosan (corresponding to 150,000 Da), which may be too high to diffuse through the cell wall. Similar effects have been reported by Tegos et al. [27] and No et al. [28], where a decrease in the antimicrobial properties was observed using CHS with molecular weight (MW) in the range of 4000 to 22,000 Da; conversely, Luksiene and Zukauskas [36] reported that molecules with an MW lower than 60,000 Da easily cross the outer membrane and encounter the inner plasma membrane.

As demonstrated with the initial singlet oxygen generation study, the effect of increasing oxygen concentration did not generate additional yield of singlet oxygen—which is reflected in Figure 5b, where no significant additional bacterial inactivation was observed

($p > 0.05$). Thus, the concentration of oxygen during these experiments did not appear to become a limiting factor for singlet oxygen production and bacterial inactivation.

According to Figure 2, the singlet oxygen production rate should have been reduced by increasing the pH of the suspension from ca. 5 to ca. 7; however, no significant effect was observed in bacterial suspensions containing NaChl and CHS-NaChl, despite their different pH values (7.3 and 5.3, respectively). We hypothesize that a more significant effect arises from the changes in outer cell structure in Gram-positive and Gram-negative bacteria.

Figure 6 demonstrated the challenge to inactivate *E. coli* using low-energy photons in the absence or presence of NaChl; however, inactivation was observed with the CHS complex. The successful photosensitized-based inactivation of Gram-positive *S. aureus* was accomplished with simply NaChl as the PS (Figure 5). A decrease in viable bacteria concentration of two orders of magnitude was attained within 105 min of irradiation with visible light. Traditionally, it is accepted that Gram-positive bacteria offer greater resistance towards ROS attacks compared to Gram-negative bacteria, since the former possess a thicker cell wall [24]; however, a different bacterial inactivation mechanism has been proposed for photosensitization. Gram-positive bacteria have a more porous cell wall, whereas Gram-negative bacteria possess a more complex cell wall structure consisting of an inner and outer membrane, the latter being made up of porins, lipoproteins, and lipopolysaccharides (LPSs), separated by a peptidoglycan layer [19,30,36]. Hence, singlet oxygen [30] or even the PSs [13,23,26,36] are likely to cross the Gram-positive bacteria cell wall and, subsequently, diffuse into the inner membrane, increasing bacterial inactivation efficiency due to the internal production of singlet oxygen. Additionally, the production of endogenous porphyrins has been reported to impact organism photosensitivity, with Gram-positive bacteria shown to have higher accumulation compared to Gram-negative bacteria [31].

### 2.3. Bacterial Growth Inhibition

Whilst a biocidal effect can be useful in industrial cleaning, the ability to prevent the proliferation of a small number of organisms on food production surfaces, food packaging, and, indeed, food products is important and critical to food hygiene with respect to transport, storage, and product shelf-life. To investigate the biostatic properties of low-level visible-light exposure of chlorin compounds, Gram-negative *E. coli* was chosen due to their resistance to inactivation. The ability of NaChl and the CHS-NaChl complex to prevent *E. coli* growth under the most favorable conditions (37 °C, 24 h) is shown in Table 1.

**Table 1.** Effect of CHS-NaChl to prevent *E. coli* growth.

| *E. coli* Growth Inhibition Test | Viable Concentration of *E. coli* (CFU·mL$^{-1}$) [c] |
|---|---|
| LB broth + *E. coli* + CHS-NaChl [a] + light [b] | $\leq$10 [e] |
| LB broth + *E. coli* + CHS-NaChl [a] + dark | $(2.0 \pm 0.4) \times 10^6$ |
| LB broth + *E. coli* + NaChl [d] + dark | $(2.0 \pm 0.8) \times 10^8$ |
| LB broth + *E. coli* + dark | $(2.0 \pm 0.7) \times 10^8$ |

[a] CHS (0.1%)-NaChl (0.00004%, ca. 0.5 μM); [b] visible irradiation (CFL lamp); [c] incubation conditions: 24 h at 37 °C; [d] NaChl (0.5 μM); and [e] bacterial detection limit.

As with classical culture methods, the concentration of viable *E. coli* following dark culturing in optimal conditions (24 h at 37 °C in an LB broth in the dark) resulted in high levels of reproduction—a cell concentration equivalent to $2.0 \pm 0.7 \times 10^8$ CFU·mL$^{-1}$, which was not impacted by exposure to visible light. No reduction in growth was observed during dark incubation in the presence of NaChl; however, 2-log reduction was observed when the CHS-NaChl complex was included in the growth media during dark incubation. Significant biocidal properties were observed with the CHS-NaChl complex; indeed, the concentration of *E. coli* was reduced to that of the detection limit of the assay—representing the prevention of growth in the order of 8-log when compared to both the light and dark

control experiments. Other studies have also reported that no regrowth was observed during the first 15 h after photoinactivation of *Salmonella* using a chlorophyllin–chitosan complex, whereas *Salmonella* treated with photoactivated chlorophyllin did show the effects of regrowth [58]. Similar effects have been reported with mold [35] and fungal organisms [33], where growth inhibition on tomato leaves and wheat sprouts during storage was reported over extended periods, confirming the photostability of chlorophyllins.

## 3. Materials and Methods

### 3.1. Photosensitizers

Several anionic photosensitizers were selected: rose bengal (RB) (organic dye, $C_{20}H_2Cl_4I_4Na_2O_5$, CAS 632-69-9, Sigma Aldrich, St. Louis, MO, USA); chlorin e6 (ce6) (organic dye, $C_{34}H_{36}N_4O_6$, CAS 19660-77-6, Santa Cruz Biotech, Dallas, TX, USA); sodium magnesium chlorophyllin (NaChl) (food additive E-140, $C_{34}H_{31}MgN_4Na_3O_6$, CAS 15203-43-7, Carl Roth, Karlsruhe, Germany); and sodium copper chlorophyllin (NaCuChl) (food additive E-141, $C_{34}H_{31}CuN_4Na_3O_6$, CAS 11006-34-1, Sigma Aldrich). All PS stock solutions were prepared with deionized water and stored at 4 °C in the dark until use.

Chitosan (CHS) (($C_6H_{11}NO_4)_n$; 150,000 Da, CAS 9012-76-4, Sigma Aldrich), a natural cationic polysaccharide with no toxic antimicrobial properties, which is accepted as a food additive [30], was used to aid the binding of anionic PSs to negatively charged exterior surfaces on the pathogens. A stock solution of CHS (1%)-NaChl (0.0004%, ca. 5.6 μM) was prepared via the addition of 1 g of CHS and 0.18 g of HCl to deionized water and, under stirring, a previously prepared stock solution at 0.05% (ca. 700 μM) NaChl was added dropwise to a final volume of 100 mL.

### 3.2. Fluorescence Measurements for the Detection of Singlet Oxygen

Singlet Oxygen Sensor Green (SOSG®, Life Technologies, Carlsbad, CA, USA), a highly specific singlet oxygen fluorescent probe [10,16,40], was used for the detection of singlet oxygen during the photoactivation of the photosensitizers. SOSG has been reported to react with singlet oxygen to form an endoperoxide complex, SOSG-EP, which emits green fluorescence [40,41,47]. Therefore, this green fluorescence response can be directly attributed to the oxidation of SOSG by generated singlet oxygen molecules.

A fresh stock solution of ca. 5 μM SOSG was prepared daily prior to experiments. SOSG was dissolved in 100 μL of methanol and kept in the dark for 10 min. The addition of deionized water produced a final SOSG working solution of 2 μM. The SOSG reaction was quantified via fluorescence assessment (excitation at 504 nm and emission at 525 nm) using 200 μL samples in a spectrofluorometer (Tecan GENios FL, Tecan, Dublin, Ireland).

### 3.3. Spectrophotometric Analysis for the Detection of a Superoxide Anion Radical

Tetrazolium salt, 2,3-Bis(2-methoxy-4-nitro-5-sulfophenyl)-2H-tetrazolium-5-carboxalinide (XTT, Sigma Aldrich), was selected as a colorimetric probe for the detection of $O_2^{-\bullet}$ [61,62] to confirm if $O_2^{-\bullet}$ was produced in addition to singlet oxygen. XTT has been widely reported to be reduced by superoxide anion radicals to form XTT formazan [61,62], permitting detection via spectroscopy.

A stock solution of 1 mM XTT was prepared in deionized water and stored at 4 °C in the dark until required. XTT working solutions were prepared at 100 μM. XTT formazan generation was assessed colorimetrically through the increase in the absorption of the solution at 470 nm using a UV–Visible spectrophotometer (Cary 50 Bio, Varian, Palo Alto, CA, USA).

### 3.4. Photosensitization

All experiments were carried out in a Pyrex glass reactor. Suspensions were exposed to solar-simulated irradiation using a 150 W arc xenon lamp (Applied Photophysics, Leatherhead, UK) placed 22 cm away from the reactor. The incident intensity was determined to be 136 W·m$^{-2}$ (measured between 200 and 800 nm using a spectral radiometer, Gemini 180,

JobinYvon Horiba, Kyoto, Tokyo). Visible radiation was provided by a commercial CFL lamp (Compact Fluorescent Lamp) (Applied Photophysics), and output irradiation was measured to be 79 W·m$^{-2}$ (from 400 to 650 nm, Figure S1).

The absorbance spectrum for each PS was measured using a UV–Visible spectrophotometer (Cary 50 Bio Varian) and a standard 10 mm pathlength quartz cuvette (Figure S1).

Experiments with which to measure the concentration of singlet oxygen produced during the photoactivation of each PS were carried out in a working volume of 20 mL of 2 μM SOSG. To determine the optimal concentration of irradiated PSs required to generate maximum single oxygen concentration, a range of PS concentrations was examined corresponding to RB, ce6, and NaChl: 0; 0.1; 0.5; and 1 μM; NaCuChl: 0; 0.01; 0.05; 0.1; 0.5; and 1 μM. The suspension was magnetically stirred to ensure effective mixing. Samples (200 μL) were collected periodically for up to 60 min and transferred to a black 96-well microplate and maintained in the dark. The evolution of singlet oxygen was followed by fluorescence measurements as described in Section 3.2.

When studying the effect of pH on the rate of singlet oxygen production, the solution pH was modified prior to irradiation by adding dropwise HCl or NaCl under stirring, with pH measured with a meter (pH 510, Eutech, Singapore). To examine the effect of solution oxygen concentration on the rate of singlet oxygen production, the working solution was air-sparged using a small aquarium pump under an open atmosphere, with a flow rate of 900 cm$^3$·min$^{-1}$, obtaining 9 mg/L at 20 °C (considered as oxygen saturation at room temperature).

### 3.5. Bacterial Growth

*Escherichia coli* (ATCC 23631) and *Staphylococcus aureus* (ATCC 29213) were used as model Gram-negative and Gram-positive bacteria, respectively, for the photoinactivation experiments. Fresh liquid cultures were prepared by the overnight inoculation of a Luria–Bertani (LB) nutrient medium for *E. coli* and Tryptic Soy Broth (TSB) for *S. aureus* from stock plates, with static incubation at 37 °C for 18–24 h. Cells were harvested by centrifugation (4000 rpm for 10 min) and washed twice with sterile ¼ strength Ringer's solution before being resuspended in the same solution as working stocks of $2 \times 10^8$ CFU mL$^{-1}$ *E. coli* and $8 \times 10^8$ CFU·mL$^{-1}$ *S. aureus*. Bacterial stock suspensions were diluted and immediately used for photosensitization experiments, as described in Section 3.6. The full details of the composition of ¼ strength Ringer's solution, components of the LB medium, and measurements of culture absorbance are available elsewhere [63].

### 3.6. Bacterial Photoinactivation

Bacterial photoinactivation experiments were carried out using the experimental equipment described in Section 3.4., via photoinactivation using (i) UV–Visible irradiation provided by the 150 W arc xenon lamp, including 2 UV-C filters (λ < 280 nm) (UQG Optics Ltd., Cambridge, UK) placed between the reactor and the irradiation source, with the incident light intensity corresponding to 105 W·m$^{-2}$ (280 to 800 nm), and (ii) visible radiation provided by a commercial CFL lamp where the intensity of the output at 405 nm matched the peak absorption wavelength of the chlorophyllin compounds (see Section 2.1.1.), corresponding to 1.32 W·m$^{-2}$.

For bacterial inactivation experiments, the stock suspension was serially diluted with ¼ strength Ringer's solution to attain an initial concentration of approximately either $1 \times 10^6$ or $1 \times 10^3$ CFU mL$^{-1}$. Appropriate concentrations of photosensitizers were added to the bacterial suspension, including NaChl and NaCuChl at 0.5 μM (pH 7.3) and the complex of CHS (0.1%)-NaChl (0.00004%) at ca. 0.5 μM (pH 5.3). The pH value of each working suspension was measured to be 7.3 in the case of both chlorophyllins, and 5.3 in the case of the CHS-NaChl complex.

Samples (200 μL) were periodically withdrawn during photoinactivation and control experiments and quantified following standard serial dilution in sterile ¼ strength Ringer's solution. Quantification was conducted via the Miles and Misra method, spotting 10 μL

drops of each decimal dilution 4 times onto LB agar plates for *E. coli,* and Tryptic Soy Agar (TSA) for *S. aureus*. Following longer irradiation times, 2 drops of 100 µL were removed from the reactor and plated directly onto either LB agar or TSA. Likewise, with experiments using an initial concentration of $1 \times 10^3$ CFU mL$^{-1}$, 100 µL drops were plated in duplicate. All plates were incubated at 37 °C for 24 h, and subsequently colonies were visually observed as well manually counted.

### 3.7. Inhibition of Bacterial Growth

To assess the potential of PSs to elicit a biostatic response, *E. coli* growth was monitored in the presence of a CHS (0.1%)-NaChl (0.00004%) complex. LB broth (10 mL) was inoculated with a sample of *E. coli* stock suspension (as prepared in Section 3.5) with the subsequent addition of the CHS-NaChl complex. The culture was irradiated with visible light (CFL lamp) throughout 24 h of incubation at 37 °C. Dark control experiments consisted of 10 mL of the same fresh bacterial culture of *E. coli* in LB broth in the presence of (i) no PS complex, simply LB broth; (ii) NaChl (0.00004%); and (iii) the complex CHS-NaChl. Following treatment, the concentration of viable bacteria was quantified according to the details given in Sections 3.5 and 3.6.

### 3.8. Statistical Analysis

Each experiment was performed in triplicate with one-way analysis of variance (ANOVA), analyzed using SPSS 20.0 (SPSS, Inc., Chicago, IL, USA). Mean values were compared via Tukey's honest significant difference (HSD), and Fisher's least significant difference (LSD) test was used to distinguish which means were significantly different from others at a statistical significance of 5% ($p \le 0.05$).

### 4. Conclusions

Food-safe chlorophyllin (NaChl, E140) has been shown under visible irradiation to yield a high rate of singlet oxygen comparable to that of rose bengal and chlorin e6 (ce6), commonly used photosensitizers in PDT with high quantum yields, but which are not approved for use in food-production environments. We demonstrate that singlet-oxygen-mediated inactivation is effective against Gram-positive bacteria (*S. aureus*) under low levels of visible light. A strong biostatic effect towards Gram-negative bacteria (*E. coli*) was observed during 24 h long experiments.

The main novelty of this study relates to further knowledge on identifying the inactivation mechanism and sensitivity of pathogens to NaChl-based photosensitization. This study contributes to the work elucidating the impact of cell wall structure (Gram-positive vs. Gram-negative bacteria) and a greater understanding of the need to generate singlet oxygen near bacteria. The impact of adding CHS to PSs can play a role in increasing the permeability of the bacterial outer membrane, resulting in enhanced levels of disinfection. Future work could be directed towards enhancing the attachment of NaChl using other biopolymers or introducing strategies to increase the permeability of the bacterial outer membrane.

With respect to the translation of the approach into food-production environments, the results demonstrate the ability to produce high rates of singlet oxygen at close to neutral pH values with the potential to simply mix food-safe compounds into water to produce a ready-to-use solution. Irradiation can be provided by visible sources, with the potential to incorporate low-cost LED arrays as bespoke UV–Vis sources. These practical considerations are important factors for applications in high-volume, low-margin food-processing industries.

The further development of NaChl and CHS-NaChl into films or spray solutions is required to aid with the translation of singlet oxygen-based photosensitization into an effective low-toxicity tool for cross-contamination prevention and microbial inactivation not only of model microorganisms but also bacterial biofilms within real food-processing environments. Further analysis by using predictive microbiology tools to develop quantita-

tive microbial risk assessment studies are required to prove the efficacy of this proposed food inactivation technology.

**Supplementary Materials:** The following supporting information can be downloaded at: https://www.mdpi.com/article/10.3390/catal14080507/s1, Figure S1: Absorption spectrum of the photosensitizer compounds studied and emission spectra for the two irradiation sources used. Figure S2: Inactivation of *E. coli* by chlorophyllin-based photosensitization in ¼ strength Ringer's solution under UV–Vis light. PS concentration: 0.5 µM.

**Author Contributions:** Conceptualization, C.P. and P.S.M.D.; methodology, C.P., P.S.M.D. and J.W.J.H.; software, C.P. and J.W.J.H.; validation, C.P. and P.S.M.D.; formal analysis, C.P. and P.S.M.D.; investigation, C.P.; resources, P.S.M.D., J.M. and R.v.G.; data curation, C.P.; writing—original draft preparation, C.P.; writing—review and editing, C.P., J.M., R.v.G., J.W.J.H., N.G.T. and P.S.M.D.; visualization, C.P.; supervision, P.S.M.D., J.M. and R.v.G.; project administration, P.S.M.D.; funding acquisition, P.S.M.D. and J.M. All authors have read and agreed to the published version of the manuscript.

**Funding:** This research was funded by the Ministry of Science and Innovation through the project AquaEnAgri (2022/00305/017) (PID2021-126400OB-C32) and the Comunidad de Madrid and Rey Juan Carlos University through the program Young Researchers R&D Project (Bio-PhLoW)-(M2727).

**Data Availability Statement:** https://zenodo.org/records/12803946. DOI: 10.5281/zenodo.12803945. Accessed on 24 July 2024.

**Acknowledgments:** The authors gratefully acknowledge the financial support of the Ministry of Science and Innovation through the project AquaEnAgri (2022/00305/017) PID2021-126400OB-C32) and the Comunidad de Madrid and Rey Juan Carlos University through the program Young Researchers R&D Project (Bio-PhLoW) (M2727).

**Conflicts of Interest:** The authors declare no conflicts of interest.

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
