# Peer review of "Assessment of Photoactivated Chlorophyllin Production of Singlet Oxygen and Inactivation of Foodborne Pathogens"

_catalysts, doi:10.3390/catal14080507_

Round 1

Reviewer 1 Report

Comments and Suggestions for Authors

The manuscript “Assessment of Photoactivated Chlorophyllin Production of Sin- 1

glet Oxygen and Inactivation of Foodborne Pathogens” reports the use of different photosensitizers (PS) for the generation of reactive oxygen species and photoinactivation of microorganisms. The work is interesting, but needs changes to be accepted.

1)      The authors report the use of different PS for the generation of singlet oxygen with the same lighting source. Therefore, it is important to show the overlap of the source emission spectrum and the PS absorption spectrum. We still have to calculate the fraction of light actually absorbed by the PS.

2)      On page 6 line from line 259, the description of the process is wrong, it is probably first order kinetics that can be adjusted to the model. It is important to note that not all kinetics reached infinity in the experiment.

3)      Normally copper chlorophyllins do not lead to the formation of singlet oxygen, as the authors explain the effect observed in Figure 1. Has the formation of singlet oxygen by this molecule been identified in a direct measurement?

4)      The experiments reported in Figure 2 do not make sense, the ideal is to calculate the quantum yield of singlet oxygen. To evaluate the effect of PS concentration, a pseudo-first order condition should be maintained, that is, the concentration of the fluorescent probe should be much higher than that of PS, which can result in many errors, such as the effect of internal filter.

5)      Regarding the effect of pH on the generation of singlet oxygen, there is a change in the species present with the variation in pH, which can alter the PS's photophysical characteristics and solubility. There is a lack of discussions and experimental characterizations in this part.

6)      In relation to figure 4, it is actually shown that RB has the lowest tendency to undergo self-aggregation. Authors should review the discussion presented.

Comments on the Quality of English Language

Good

Author Response

We wish to thank all reviewers for their comments, which we acknowledge have significantly improved the quality of the paper.

REVIEWER 1

The authors report the use of different PS for the generation of singlet oxygen with the same lighting source. Therefore, it is important to show the overlap of the source emission spectrum and the PS absorption spectrum. We still have to calculate the fraction of light actually absorbed by the PS.

The reviewer is correct to highlight this important point. This overlap between emission spectrum and the PS absorption spectrum is reported in Figure S1 (Supplementary material)

The experiments reported in Figure 2 do not make sense, the ideal is to calculate the quantum yield of singlet oxygen. To evaluate the effect of PS concentration, a pseudo-first order condition should be maintained, that is, the concentration of the fluorescent probe should be much higher than that of PS, which can result in many errors, such as the effect of internal filter.

The reviewer is again correct, and we thank them for this helpful comment.  PS concentrations of 5 and 10 µM should not appear in the figure, in addition we have removed data for 1 µM as it may still be a high concentration considering the similar initial concentration to 2 µM SOSG.  We have added a specific comment relating to the limitations of the assay used in this study

On page 6 line from line 259, the description of the process is wrong, it is probably first order kinetics that can be adjusted to the model. It is important to note that not all kinetics reached infinity in the experiment.

The reviewer is right. We reached a plateau when using higher PS concentrations (1, 5, 10 µM) than that of the SOSG, where there was potential for the SOSG concentration to be a limiting factor.  In updating the data we used a zero-order kinetic model to calculate kinetic constants from 0.05, 0.1, and 0.5 µM fitting points from 0 to 16 min based on the initial rate method considering low initial conversions.  From this analysis we should have observed a linear increase in kinetic constants (singlet oxygen formation) with increasing PS concentration, but we did not see this effect. Thus, we are unfortunately not able to calculate the apparent quantum yield, but include commentary on this area within the revised manuscript.

During our consideration of the reviewers comments, we have decided to remove Figure 2 from the manuscript.  We now refer only to zero-order kinetic values of 0.5 µM PS concentration. Our main goal of this section of the work was only to confirm singlet oxygen production for a low PS concentration and establish a practical approach to develop a sustainable method of pathogen inactivation for the food industry.  We draw comparison between the singlet oxygen production of NaChl  and well recognised PS molecules such as RB and ce6 – which due to toxicity cannot be used in the area of application in which the paper is focused.

Our decision on the updated kinetic approach is also informed by literature, where X. Xie, Z. Zhang, Y. Hu, H. Cheng, , Chemical Engineering Journal, 334, 2018, 1242-1251, describe that both pseudo-first- and pseudo-second-order kinetic models could fail to accurately describe the degradation rates of substrates in self-sensitized photodegradation.  We agree that there is a significant need to develop a mechanistic based kinetic model to fully optimize these PS systems and resolve the mystery surrounding both the kinetics and reliably to predict substrate photodegradation rate under environmentally relevant conditions - but acknowledge that this important task was not the aim of this work or the goal of the Special Issue to which we have submitted the paper.

Updated section in the paper now reads:

The effect of the PS concentration in the range 0.01-1.0 µM on singlet oxygen generation was measured with a fixed concentration of SOSG in excess (2 µM). The zero-order kinetic constants were calculated from the initial reaction rate of time-dependent fluorescence enhancement of SOSG i.e. the concentration of SOSG-EP. As reported in literature, 1O2 production rates are often observed to increase linearly as a function of the initial PS concentration, however, in this case we cannot report that expected effect in the narrow range of studied PS concentrations (data not shown). Within the range of concentrations evaluated, we obtained maximum singlet oxygen formation at 0.5 µM, but recognize the potential for rate limitation at 1 µM of PS due to the fixed concentration of SOSG (2 µM). As such, 0.5 µM was considered as an appropriate PS concentration for further experimentation, effectively using the incident radiation to produce consistent levels of singlet oxygen. 

Both chlorophyllin based compounds exhibited singlet oxygen production under ‘practical’ conditions, whereas literature values are typically reports the use of concentration values ranging from 10-150 µM and exposure to higher light intensities [26, 37, 46, 47].  While the initial 1O2 rate constant for NaChl (293±22 µM·min-1) was similar (p > 0.05) to those obtained for RB and ce6, 323 and 291±21 min-1, respectively, NaCuChl showed a lower value of 143±23 min-1 (p ≤ 0.05), demonstrating significant potential for the use of NaChl as an effective food-safe biocidal compound. We may have expected the highest singlet oxygen production for RB and ce6 with values of reported quantum yield of 0.75 and 0.7 respectively [21, 45] followed by NaChl with values of reported quantum yield of 0.39. Low yields of singlet oxygen production would be consistent with reported quantum yields for NaCuChl of 0.03 [45].

Normally copper chlorophyllins do not lead to the formation of singlet oxygen, as the authors explain the effect observed in Figure 1. Has the formation of singlet oxygen by this molecule been identified in a direct measurement?

Due to equipment limitation we have not been able to directly confirm the formation of singlet oxygen with any other probes or measurements. We have however carefully and consistently run all the control experiments (dark, no PS, SOSG; light, no PS, SOSG; dark, PS, SOSG) and conducted multiple replicates over many months to rule out any potential singlet oxygen formation apart from that generated from the photo excitation of the PS molecules. Also, we have run tests with XXT, as indicated in the text, to rule out superoxide ion formation. If other ROS had been formed, we would expect to see the associated biological effects in the microbial inactivation, which we did not observe during the disinfection studies.

.

Regarding the effect of pH on the generation of singlet oxygen, there is a change in the species present with the variation in pH, which can alter the PS's photophysical characteristics and solubility. There is a lack of discussions and experimental characterizations in this part.

As our aim of this element of the paper was to determine if different pH values would limit singlet oxygen production – and as such confirm that ‘special’ or ‘complex’ or with respect to food production, ‘unsafe’ conditions were not required to use this approach in a high through-put industrial environment (see last paragraph of the introduction). As with a very detailed mechanistic study, we did not determine the chemical PS structures formed at different pH or indeed study aggregation of PS molecules leading to reduced singlet oxygen production - but we absolutely recognize the importance of the reviewers comment and have included further discussion of these points.

Updated section in the paper now reads: RB potential to produce singlet oxygen was only affected at low pH values, in agreement with Neckers [49], where acid conditions result in formation of an ester group causing ring closure. Thus, at a pH range of 2.7-3.2, an important shift occurs in the absorption maximum of RB. When pH is less acidic, the open ring RB structure dominates [50]. Moreover, Lin et al. [39] noted the reaction between SOSG and singlet oxygen to be sensitive to the solution pH, temperature, dissolved oxygen concentration, and as such further investigation could be performed to supplement use of this indirect 1O2 probe in specific microenvironments.  From a practical perspective, the data confirm that a general supply of distilled water (slightly acidic) would be sufficient for the preparation of a functional chlorophyllin solution for industrial application.

In relation to figure 4, it is actually shown that RB has the lowest tendency to undergo self-aggregation. Authors should review the discussion presented.

Again, we thank the reviewer for highlighting this point, but suggest that a fundamental mechanistic analysis is unfortunately beyond the scope of the very applied (practical) focus of this study.  We do recognize that PS molecule aggregation can impact singlet oxygen production, in particular where PS molecules have low solubility, but also acknowledge that research over the past 20 years has harnessed this effect to play a positive role through aggregation-induced emission.  We understand that RB and RB-based molecules can play important roles in these processes and in accordance with many other interests in this research, we are working to secure funding to further investigate the application of these well studied PDT effects for use in food-processing environments, subject to restrictions related to food safety and cost.  We have added a comment to the paper to recognize aggregation (agglomeration) where impact may be more pronounced for PS molecules other than RB.

Updated section in the paper now reads:

It is generally accepted that the presence of oxygen is necessary in the system to yield singlet oxygen [40, 48, 50-53] specifically in Type II reactions [54], however full oxygen saturation is not required and the additional sparging could induce PS agglomeration impacting RB to a lesser extent than the other PS molecules.

Reviewer 2 Report

Comments and Suggestions for Authors

This paper describes photosensitized production of singlet oxygen by chlorophyllin complexes for inactivation of the foodborne pathogens. The problem addressed in this paper is relevant, and the advantages of the solution proposed include food safety, low cost, and potential to overcome antimicrobial resistance. A comprehensive introduction provides sufficient information on the relevance of APDT application in food processing environments. Materials and method are described in details. The results obtained are well illustrated and self-consistent. Particularly, the authors compared singlet oxygen generation efficiency for different photosensitizers at different pH and oxygen concentrations, as well as their biocidal activity under UV-vis and visible light irradiation. Of special significance are the different mechanisms proposed for the chlorophyllin complex interactions with Gram-positive and Gram-negative bacteria. As a whole, conclusions made by the authors are well grounded and convincing, so the paper is recommended for publication after minor revision.

- Before considering chlorophyllin-based photosensitizers suitable for APDT application in food processing industry, the authors should inevitably address photostability of the chlorophyllin complexes upon irradiation, since they are known to be highly subjected to autosensitized photobleaching, especially under prolonged UV-Vis light irradiation.

- The reference list includes about 13 papers by Luksiene et al., which is more than 20% of the total references. Despite the absence of self-citations, it is strongly recommended to replace some of the above references with the works performed by the other groups.

Author Response

We wish to thank all reviewers for their comments, which we acknowledge have significantly improved the quality of the paper

REVIEWER 2

Before considering chlorophyllin-based photosensitizers suitable for APDT application in food processing industry, the authors should inevitably address photostability of the chlorophyllin complexes upon irradiation, since they are known to be highly subjected to autosensitized photobleaching, especially under prolonged UV-Vis light irradiation.

The reviewer is right, we refer to that point in the text when saying: Sustained production of high concentrations of singlet oxygen from all PS molecules was observed for at least 60 min.

We think photostability is a key point when increasing temperatures (Marvin et al. 1999), however in the food industry where we are working at low temperature and with visible light irradiation we do not foresee an issue. To help provide the reviewer with confidence on this point, we include date which confirmed the inhibition in E. coli growth in presence of CHS-NaChCl during visible light for 24 h validating the activity of the PS material over that extended period of time. Josewin and Rodríguez-López et al. also confirm prevention of growth for 48 and 16 h respectively agreeing with the data presented in this paper.

Marvin L. Salin, Luis M. Alvarez, Bert C. Lynn, Bahanu Habulihaz & Augustus W. Fountain iii (1999) Photooxidative Bleaching of Chlorophyllin, Free Radical Research, 31:sup1, 97-105.

W. Josewin, M.-J. Kim, H.-G.Yuk. Inactivation of Listeria monocytogenes and Salmonella spp. on cantaloupe rinds by blue light emitting diodes (LEDs). Food Microbiology 76 (2018) 219–225.

The reference list includes about 13 papers by Luksiene et al., which is more than 20% of the total references. Despite the absence of self-citations, it is strongly recommended to replace some of the above references with the works performed by the other group

We acknowledge this as an important point and have removed some references to Luksiene et al. where the authors work overlaps with other sources, such as: [18-20], [23], [26], [28] and we have supplemented the discussion with some modern references from a range of authors - including suggestions from Reviewer 3, as shown below.

Jiang, F. Scholle, F. Jin, Q. Wei, Q. Wang, R. A. Ghilad Chlorophyllin as a photosensitizer in photodynamic antimicrobial materials. Cellulose (2024) 31:2475–2491

Srivastava, P. K. Singh, A.Ali, P. P. Singh, V. Srivastava. Recent applications of Rose Bengal

catalysis in N-heterocycles: a short review. RSC Adv., 2020, 10, 39495

P. Singh, S. Sinha, P. Gahtori, D.N. Mishra, G. Pandey, V. Srivastava. Recent advancement in photosensitizers for photodynamic therapy. Dyes and Pigments 229 (2024) 112262

  1. Srivastava, P. K. Singh, P. P. Singh. Recent advances of visible-light photocatalysis in the functionalization of organic compounds. Journal of Photochemistry & Photobiology, C: Photochemistry Reviews 50 (2022) 100488

  1. W. Josewin, M.-J. Kim, H.-G.Yuk. Inactivation of Listeria monocytogenes and Salmonella spp. on cantaloupe rinds by blue light emitting diodes (LEDs). Food Microbiology 76 (2018) 219–225.

  1. C. Neckers. (1988) Rose Bengal review, Journal of photochemistry and photobiology, A: Chemistry 1–29.88

Amat-Guerri, Lopez-Gonzalez, M. M. C., Martinez-Utrilla, R., and Sastre, R. (1990) Synthesis and spectroscopic properties of new rose bengal and eosin Y derivatives, Dyes and pigments 12.

Reviewer 3 Report

Comments and Suggestions for Authors

Recommendation: Minor revision

The claim of the manuscript is "Assessment of Photoactivated Chlorophyllin Production of Singlet Oxygen and Inactivation of Foodborne Pathogens" by Marugán et al. which is a well-chosen topic. I would like to recommend this work to ‘Catalysts’ after the following minor revision.

Comments:

1. Give the journal mechanism of photocatalytic action.

2. Cite this review in introduction section Journal of Photochemistry and Photobiology C: Photochemistry Reviews, 50, 2022, 100488; RSC Adv., 2020, 10, 39495; Dyes and Pigments 229 (2024) 112262.

Author Response

We wish to thank all reviewers for their comments, which we acknowledge have significantly improved the quality of the paper

Give the journal mechanism of photocatalytic action. Cite this review in introduction section Journal of Photochemistry and Photobiology C: Photochemistry Reviews, 50, 2022, 100488; RSC Adv., 2020, 10, 39495; Dyes and Pigments 229 (2024) 112262

In line with this comment, and those from previous reviewers, we have revised and updated to section on mechanism and photocatalytic action.  We have also included the three references within the introduction.

P. Singh, S. Sinha, P. Gahtori, D.N. Mishra, G. Pandey, V. Srivastava. Recent advancement in photosensitizers for photodynamic therapy. Dyes and Pigments 229 (2024) 112262

Srivastava, P. K. Singh, A.Ali, P. P. Singh, V. Srivastava. Recent applications of Rose Bengal catalysis in N-heterocycles: a short review. RSC Adv., 2020, 10, 39495

Srivastava, P. K. Singh, P. P. Singh. Recent advances of visible-light photocatalysis in the functionalization of organic compounds. Journal of Photochemistry & Photobiology, C: Photochemistry Reviews 50 (2022) 100488

Reviewer 4 Report

Comments and Suggestions for Authors

This work aimed to evaluate the biocidal and biostatic effects of food-safe chlorophyllin compounds for application in food processing environments using low-energy photon sources. To ensure ideal conditions for high levels of pathogen inactivation, the project aimed to gain an understanding of the environment required to generate a high concentration of 1O2  using the Singlet Oxygen Sensor Green fluorescence probe to detect 1O2  with minimal requirement for preparation, use of expensive reagents or complex approaches.

 The paper will be useful as a reference for scientists who employ photoactivated chlorophyllin compounds and foodborne pathogens for a wide range of applications.   Overall, the quality, quantity, and novelty of the work justify the publication the paper is acceptable in its current form.

Author Response

No changes required

Round 2

Reviewer 1 Report

Comments and Suggestions for Authors

The article can be accepted.